# A Comparison between Elite Swimmers and Kayakers on Upper Body Push and Pull Strength and Power Performance

**DOI:** 10.3390/ijerph17228301

**Published:** 2020-11-10

**Authors:** Sandro Bartolomei, Giorgio Gatta, Matteo Cortesi

**Affiliations:** 1Department of Biomedical and Neuromotor Sciences, University of Bologna, 40138 Bologna, Italy; 2Department for Life Quality Studies, University of Bologna, 47921 Rimini, Italy; giorgio.gatta@unibo.it (G.G.); m.cortesi@unibo.it (M.C.)

**Keywords:** prone bench pull, bench press, assessments

## Abstract

The aim of the present study was to compare the load-power curve expressed at bench press (PR) and prone bench pull (PU) between elite swimmers and kayakers. Another aim was to calculate the strength and power PR/PU ratio in the same populations. Fifteen elite swimmers (SW: age = 23.8 ± 2.9 y; body mass = 82.8 ± 5.6 kg; body height = 184.1 ± 4.6 cm) and 13 elite kayakers (KA: age = 23.8 ± 2.9 y; body mass = 91.0 ± 3.5 kg; body height = 180.1 ± 5.4 cm) were assessed for PR 1RM and PU 1RM. They were then assessed for power produced at 40, 60 and 80% of 1RM in both PR and PU. The area under the load-power curve (AUC) and PR/PU ratios were calculated for both the SW and KA groups. The KA group showed significantly higher PR1RM (+18.2%; *p* = 0.002) and PU1RM (+25.7%; *p* < 0.001) compared to the SW group. Significant group differences were also detected for PUAUC (*p* < 0.001) and for the PR/PU power ratio (*p* < 0.001). No significant group differences were detected for PRAUC (*p* = 0.605) and for the PR/PU strength ratio (*p* = 0.065; 0.87 and 0.82 in SW and KA, respectively). The present findings indicate that elite KA were stronger and more powerful than elite SW in the upper body. Not consistently with other athletic populations, both KA and SW athletes were stronger and more powerful in upper body pull compared to push moves.

## 1. Introduction

Dryland resistance exercises are widely used to develop training adaptations in athletes competing in various water sports, including swimming, rowing and kayaking [1,2]. Recently, resistance exercises, such as bench press, lateral pull down, pull ups and prone barbell row, have been extensively used to assess upper body pressing and pulling strength and power in recreationally active adults [3], and in different athletic populations including competitive kayakers [4,5,6]. Upper body pressing and pulling maximal strength showed large correlations with flat-water sprint kayak performance in elite athletes [7]. Both pull and push moves have been identified as determinant for the mechanics of the kayak stroke [8]. In swimming, scientific investigations on the relationship between dryland push and pull strength and power assessments and swimming performance came to incongruous conclusions [1,9,10]. More recently, however, large correlations have been detected between the power produced at the bench press and the average force produced in tethered swimming by well-trained swimmers [11]. Consistently with these findings, Perez-Olea et al. [2] supported the possibility to predict short-distance swimming performances of well-trained swimmers according to the pull-up power.

In this scenario, bench press (PR) and prone bench pull (PU) represent two of the most common exercises for both training and testing purposes among athletes competing in water disciplines [2,12]. Different typical strength and power expressions have been identified in PR compared to PU, with the first producing greater maximal forces and the latter resulting in greater speed and power [6,13]. In addition, push and pull exercises have been used to calculate strength ratios between anterior and posterior shoulder muscle groups. A good balance between opposing muscle groups may indeed improve physical performance and limit the likelihood of injury [4,14,15]. Baker and Newton [4] found a ratio of 0.98 between upper body push and pull 1 repetition maximum (1RM) in resistance trained rugby players. Similarly, ratios of 0.96 and 1.12 between bench press 1RM and bench pull 1RM were detected in elite Turkish kayakers and in competitive athletes of various disciplines, respectively [6,16]. 

Despite the studies that investigated the push and pull maximum force in different athletes, to date, no studies analyzed the power ratios in elite athletes. In addition, no studies analyzed the push–pull power ratio among elite kayakers and swimmers. This parameter may provide additional information about the relationships between anterior and posterior shoulder muscle groups from a more sport-specific perspective. Both kayakers and swimmers require a high level of power and use similar muscle groups for propulsion in water. However, the difference in drag between the boat-paddle system and the swimmer’s body requires different power expressions. Elite athletes are very specialized; however, the inclusion of PR and PU exercises in specific resistance training programs is very common among both kayakers and swimmers. Thus, the aim of the present investigation was to compare the strength–power curves at PR and PU between elite swimmers and kayakers. Another aim of the present investigation was to compare the strength and power PR/PU ratio between the two groups. It was hypothesized that different levels of maximum strength, power and strength and power ratios may exist between the two groups of athletes.

## 2. Materials and Methods

### 2.1. Participants

Twenty-eight elite male athletes volunteered to participated in the present investigation. Within the group, 15 athletes were elite swimmers (SW: Age = 23.8 ± 2.9 y; body mass = 82.8 ± 5.6 kg; body height = 184.1 ± 4.6 cm; BMI = 24.4 ± 2.5) competing in front-crawl short- and middle- distance events (100–400 m) and 13 athletes were elite sprint kayakers (KA: age = 25.7 ± 3.7 y; body mass = 83.2 ± 6.8 kg; body height = 183.1 ± 5.4 cm; BMI = 24.8 ± 2.7) competing in K1, K2 and K4 short- and middle- distance events (200–1000 m). The average world ranking of athletes in the KA group was 29.5 ± 12.3. The FINA points of athletes in the SW group were 812.1 ± 76. Participants of both groups were members of the Italian Senior National Team and selected for the training camps of their federation. Inclusion criteria required participants to be between the ages of 18 and 35 years and have competed at the international senior level in the 12 months prior to the tests. In addition, participants were required to be involved in resistance training at least three times per week and in sport-specific training at least five times per week in the three years previous to the investigation. Exclusion criteria included injuries that occurred in the year before the study. Participants were asked to abstain from alcohol, caffeine, and resistance training for at least 48 h before the tests. The study was approved by the University of Bologna Institutional Review Board. Testing procedures were fully explained to each participant before obtaining individual written informed consent.

### 2.2. Design and Methodologies

Each participant was tested in two sessions separated by 48 h. In the first session, following basic anthropometric assessments, one repetition maximum (1RM) tests were performed at bench press (PR) and at prone bench pull (PU). After 48 h without resistance training, participants performed power tests at both bench press and prone bench pull using different percentages of the previously calculated 1RM. During the first and the second visits, the exercise order was randomized for each participant by flipping a coin. The estimated sample size was 24 to detect a between-group difference of 100 w in both PR and PU.

### 2.3. Strength and Power Testing

Prior to PR and PU 1RM testing, participants performed a standardized warm-up consisting of 5 min on a cycle ergometer against a light resistance, 10 bodyweight squats, 10 bodyweight walking lunges, 10 dynamic walking hamstring stretches, and 10 dynamic walking quadriceps stretches [17]. The 1-RM tests for the barbell PR and for the PU were performed using methods previously described by Bartolomei and colleagues [18]. Briefly, each participant performed two warm-up sets using a resistance of approximately 40–60% and 60–80% of his perceived maximum, respectively. For each exercise, 3–4 subsequent trials were performed to determine the 1-RM. A 3–5 min rest period was provided between each trial. Trials not meeting the range of motion criteria for each exercise or where technique was not appropriate were discarded.

During the second visit, following the same standardized warm-up described above, participants were required to perform a bench press power test and a prone bench pull power test. In the PR power test, participants laid down on a bench in supine position with the bar racked on their chest and were instructed to push the barbell as explosively as possible until complete extension of the arms. The PU power test was performed as previously described by Lum and Aziz [5]. Briefly, participants were asked to lie prone on a high bench, grasp the barbell with hands slightly wider than shoulder width apart and with elbows fully extended. Then, they were asked to pull the bar with maximum effort until they reached the underside of the bench. Grip width was measured for each participant and the same distance was reproduced for PR and PU. In both exercises, only the concentric phase was analyzed.

Participants pressed or pulled loads corresponding to 40% (PR40 and PU40), 60% (PR60 and PU60) and 80% (PR80 and PU80) of their previously assessed PR and PU 1RM, respectively. Two trials were performed for each load with a recovery time of 3 min. During all repetitions, an optical encoder (Tendo Unit model V104, Tendo Sports Machines, Trencin, Slovak Republic) measured the mean power expressed by the participants. The highest value of power exerted by each participant in both PR and PU was registered as power peak (PPeak). Then, the area under the load-power curve (AUC) was calculated for both PR and PU using a standard trapezoidal method [19,20]. Intraclass coefficients were 0.96 (SEM: 17.5 w) and 0.97 (SEM: 16.2 w) for PR and PU PPeak, respectively.

Ratios between PR and PU 1RM were calculated as follows: 1RM ratio = PR 1RM/PU 1RM

Ratios between AUC at PR and PU were calculated as follows: P ratio = AUC PR/AUC PU. During all strength and power measurements, participants were verbally encouraged by the study investigators and all the assessment sessions were supervised by certified strength coaches. Both assessment sessions were performed at the same time of day (3 PM), using regular Olympic barbells and plates (Eleiko International, Halmstad, Sweden). All of the assessments in both the KA and SW groups were conducted during the preparatory phase, when athletes were not competing.

### 2.4. Statistics

A Shapiro–Wilk test was used to assess the normal distribution of the data. Performance data and anthropometric parameters in the two groups were compared using independent sample *t* tests.

Where appropriate, group differences were calculated as follows: [(group A mean − group B mean)/group B mean] × 100. Significance was accepted at an alpha level of *p* ≤ 0.05, and all data are reported as mean ± SD.

## 3. Results

No significant group differences (*p* > 0.05) were detected for any anthropometric parameter. All of the data resulting from strength and power assessments are reported in Table 1.

Significant differences between the SW group and the KA group were detected for PR 1RM (*p* = 0.002) and for PU 1RM (*p* < 0.001). The KA group performed 18.2% and 25.7% better than the SW group in PR 1RM and PU 1RM, respectively.

Significant differences between the groups were detected for PU PPeak (*p* < 0.001), with the KA group performing 43.6% better than SW group. Significant differences between the groups were also detected for PU40 (*p* = 0.004), PU60 (*p* < 0.001) and PU 80 (*p* = 0.022).

A significant difference between the groups was also found for PUAUC (*p* < 0.001), with the KA group performing 85.9% better than the SW group. Load–power curves of both groups for PR and PU are depicted in Figure 1 and Figure 2, respectively.

The P ratio (Figure 3) also showed a significant difference between the groups (*p* < 0.001).

No significant group differences were detected for PRAUC (*p* = 0.605), PR PPeak (*p* = 0.919), PR40 (*p* = 0.108), PR60 (*p* = 0.202), PR80 (*p* = 0.852) and 1RM ratio (*p* = 0.065).

## 4. Discussion

The present study compared elite KA and SW athletes on maximum strength and power produced at PR and PU exercises. Significant differences between the groups were detected in both exercises, with the KA group being stronger than the SW group by 18.2% and 25.7% in PR and PU, respectively. Despite KA being significantly stronger than SW in both PR and PU, no significant differences between the two groups were detected for the 1RM ratio. The 1RM ratios of both KA (0.82) and SW (0.87) groups indicate higher levels of maximum strength produced by the back muscles compared to the anterior muscles (PR). This is not consistent with findings of other investigations performed on elite athletes (1.12) [6] and on rugby players (0.98) [4]. Electromyographic studies demonstrated that back muscles, and in particular the latissimum dorsi, are the primary muscles of propulsion in freestyle swimming [21]. Consistently, latissimum dorsi also has a major role in the propulsive phase of a kayak stroke [22]. McKean et al. [7,23], however, reported a similar 1RM ratio compared to the present study in elite KA (0.76). The authors suggest that coaches should consider identifying this push–pull ratio in sprint kayak paddlers as part of the specific training program [7]. Changes of 10 kg in 1RM pull have been associated with changes of 1% in KA performance in male elite athletes. Curiously, a push–pull ratio of 0.98 was found in elite Turkish kayakers. Turkish kayakers, however, may not represent the world elite. On the contrary, in a study by McKean [7] and in the present investigation participants were top-level world-class athletes.

Significantly higher power performances were registered in PU in the KA group. On the contrary, both AUC and PPeak produced at PR were not significantly different between the two groups. Power represents a crucial factor in both sprint swimming and kayaking. The biomechanical differences between these disciplines, however, may shift the physiological needs of KA athletes toward a higher requirement of maximum strength compared to SW. The need for high levels of maximal strength and power in KA mainly involves shoulder back muscles. Indeed, the PU 1RM values detected in both the KA and SW groups are similar compared to other high-level strength and power athletes, such as professional rugby players [4]. On the contrary, PR 1RM values in both the KA and SW group were lower compared to the aforementioned population of strength and power athletes (142.7 kg in professional rugby players vs. 108.7 in KA and 92.1 in SW) [4]. In addition, Baker and Newton [4] reported an average body mass of 94.4 kg in professional rugby players while the average body mass of both KA and SW participating in the present study was about 83 kg. KA and SW athletes may indeed be stronger than other athletic populations in pull strength relative to body mass. As recently reported by Sanchez-Medina et al. [6], PR and PU are characterized by different power–load relationships. In particular, higher velocities and power were observed in PU, while greater values of maximum strength were observed in PR [6]. Higher PPeak values were also detected in the present investigation in PU compared to PR, while maximum strength was significantly higher in PU compared to PR in both the KA and SW groups. Resistance training programs, as well as sport-specific workouts, in both KA and SW may be more focused on pulling than on pushing movements in the attempt to target specific propulsive muscle groups.

All of the participants in the present study obtained the PPeak at 60% of 1RM in both PR and PU. This is consistent with previous investigations indicating that the power output was maximized for loads between 40 and 70% of 1RM for PR and between 50 and 90% for PU [6]. These authors, however, measured power produced during the propulsive phase of both PR and PU using loads from 30 to 100% of 1RM, in 5% increments. Contrariwise, in the present study, only three loads (40–60–80% of 1RM) were used for power assessments and that may not be enough to detect the exact load that maximizes power production.

A significant difference in the power ratio was detected between KA and SW, with the KA group showing a significantly lower ratio between push and pull power compared to the SW group. The KA group showed a clear prevalence of back muscle power over front muscle power. Higher levels of power detected in PU in the KA group compared to the SW group demonstrate that KA is a more power-oriented sport compared to SW. Despite muscular power appearing to be an important factor for sprint swimming performance, propelling efficiency and water perception seem to be key factors for swimming performance [24]. Contrariwise, in KA, the paddler is required to generate a high power during each stroke to maximize kayak speed. At an average boat velocity of 4.6–5.4 m/s, the paddle stays in the water for about 0.47 s and the pull phase should be completed in a minimal amount of time [25]. Thus, KA represents a high-speed power-oriented discipline. The lower velocity of the swimmer is partially compensated for by the absence of paddles, and thus the shoulder joint’s angular velocity may not be different between sprint swimming and kayaking. The absence of paddles, however, may reduce the need for maximal pull strength in swimming.

Shoulder muscle imbalances between agonist and antagonist has been associated with changes in anthrokinematics and joint movement patterns that may lead to inflammation and shoulder pathologies, such as the subacromial impingement syndrome [26]. Baker and Newton [4] suggested that the ratio between upper body pressing and pulling strength should be approximately 1. Contrariwise, resistance-training programs are often more focused on large muscles, such as pectorals and deltoids, than on the rotator cuff muscles involved in shoulder stability [27].

Thus, both KA and SW athletes participating in the present study had a good balanced between contrasting muscles of the shoulder girdle. Previous investigations reported that 53% of elite international KA have experienced shoulder injuries [28]. Future studies may investigate if shoulder injuries in elite athletes are related to transitory changes in strength and power ratios between agonist and antagonist muscles.

## 5. Conclusions

The findings of the present investigation indicate that elite KA athletes are characterized by higher levels of upper body pull strength and power compared to elite SW sprinters. Contrariwise, no significant differences in upper body pushing strength and power were detected between KA and SW athletes. Interestingly, both KA and SW athletes were stronger and more powerful in upper body pull compared to push moves. To the best of our knowledge, this is the first study to investigate the ratio between push and pull power in elite athletes (P ratio). KA athletes showed a push–pull P ratio of 0.50 while SW showed a ratio of 0.89. A dominance of upper body pull on push power characterizes both KA and SW; in KA, however, pull power may represent a more important variable than in SW.

This suggest that each sport may have its own unique push–pull strength and power ratio.

## Figures and Tables

**Figure 1 ijerph-17-08301-f001:**
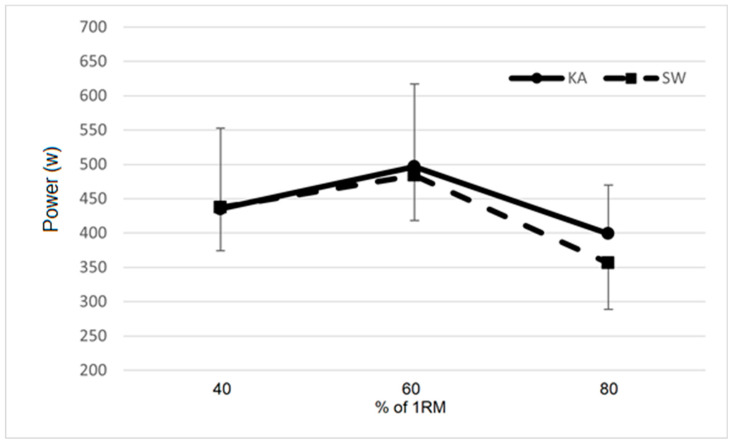
Load–power curve expressed at bench press (PR) by the KA and SW groups. KA = kayak group; SW = swimming group.

**Figure 2 ijerph-17-08301-f002:**
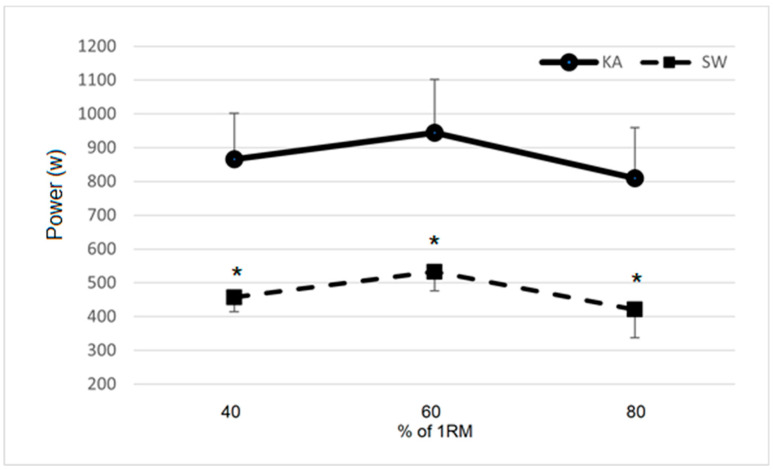
Load–power curve produced at prone bench pull (PU) by the KA and SW groups. KA = kayak group; SW = swimming group; a.u. = arbitrary units. * indicates a significant (*p* < 0.05) difference between the groups.

**Figure 3 ijerph-17-08301-f003:**
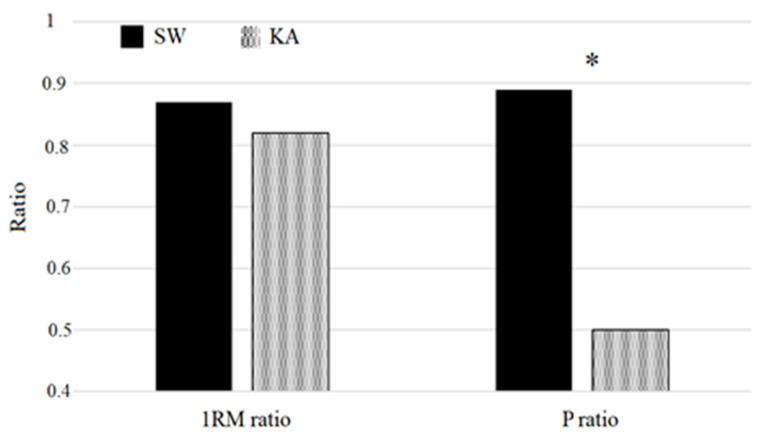
Maximum strength (1RM) and P ratio in the KA and SW groups. 1RM ratio = 1RM PR/1RM PU; P ratio = PRAUC/PUAUC. KA = kayak group; SW = swimming group. * indicates a significant difference (*p* < 0.05) between the groups.

**Table 1 ijerph-17-08301-t001:** Performance at maximum strength and power assessments in SW and KA. 1RM = 1 repetition maximum; AUC = area under the load-power curve; P ratio = power ratio. KA = kayak group; SW = swimming group.

Assessment	Group	SW	KA	Group Comparison (*p*)
1RM (kg)	PR	92.1 ± 11.3	108.7 ± 14.1	0.002
	PU	105.1 ± 8.8	132.0 ± 10.7	<0.001
AUC (a.u.)	PR	75,049.2 ± 10,805.4	79,145.3 ± 16,475.8	0.605
	PU	83,907.7 ± 10,263.3	156,045.3 ± 25,498.8	<0.001
PPeak (w)	PR	484.0 ± 65.8	501.3 ± 118.1	0.919
	PU	532.8 ± 56.8	952.2 ± 153.1	<0.001
1RM ratio		0.87 ± 0.06	0.82 ± 0.08	0.065
P ratio		0.89 ± 0.08	0.50 ± 0.06	<0.001

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
