# Peer review of "A Comparison between Elite Swimmers and Kayakers on Upper Body Push and Pull Strength and Power Performance"

_ijerph, 2020, doi:10.3390/ijerph17228301_

Round 1

Reviewer 1 Report

The present study aimed to investigate elite swimmers and kayakers force-power curve. I did not find any novelty in this study, as both swimmers and kayakers are closely related. For example, in the UK, many kayakers were former professional swimmers. However, I would like to suggest a major correction, but hopefully, it can be still able to publish.

  • Abstract: many acronyms are without the actual full word. Like BP? Is it bench press? And why in line one bench press is PR? It is confusing. Then, in line 6, it is BP1RM, is it Bench press 1 RM? What is PowP?
  • Introduction, what is the rationale of the study? The readers know there is a lack of studies conducted between swimmers and kayakers, but why need to conduct this study? Please include in the introduction. The conceptual framework needs to specify more clearly. Unfortunately, the current framework of this study is unclear.
  • Methods – please include sample size calculation.
  • What events are the elite swimmers competing? Same for kayakers?
  • The area under the curve (AUC) in the abstract, but in Methods author used “area under the force-power curve (AUC)”
  • In method: tested 2 sessions separated by 48 hours. Please support with reference, is it sufficient to fully rest? How was the randomization between PR and PU occur?
  • The study involves human participants, please attached the Human ethics certificate and code.
  • Why did researchers not use better measurement like the isokinetic machine to measure power? Please cite the procedure.
  • Section “2.4 statistics” is confusing. “if the assumption of sphericity was violated…..”. This sentence is missing the main statistical analysis. Did researchers use mixed factorial ANOVA or repeated measured ANOVA? Assumption of sphericity normally checked under this Analysis (Repeated measured). The statistical software used in this analysis is not mentioned in the text.
  • Is there any interaction effect and time effect (40, 60, 80)? This was not assessed in the study.
  • What is the demographic information of the participants? Please include in a Table and analyze the differences, rather than just descriptive in the Participants section. Besides, suggest adding BMI and years of experiences competing at the international level. Is it junior international or senior? Much unknown information.
  • Similar to Table 1, suggest to analyze the difference and show the effect size and p-values between swimmers and kayakers.
  • Sampling method and study design were not mentioned in the manuscript.
  • Reference: Only one 2020, and two 2018, others are fairly old references. Besides, please include doi.
  • The authors' contribution was also imbalance, seemed like S.B did the entire experiment and writing.
  • I think this manuscript is not suitable for IJERPH, suggest submitting it under “Sports” MDPI journal.

Unfortunately, the manuscript is not ready for publication in this form.  

Author Response

The authors would like to thank the reviewers for their time and for their suggestions in the attempt to improve the overall quality of the manuscript. In the manuscript, changes have been written in red while removed periods have been highlighted in yellow.

Rev 1

The present study aimed to investigate elite swimmers and kayakers force-power curve. I did not find any novelty in this study, as both swimmers and kayakers are closely related. For example, in the UK, many kayakers were former professional swimmers. However, I would like to suggest a major correction, but hopefully, it can be still able to publish.

  • Abstract: many acronyms are without the actual full word. Like BP? Is it bench press? And why in line one bench press is PR? It is confusing. Then, in line 6, it is BP1RM, is it Bench press 1 RM? What is PowP?

Amended. BP has been replaced with PR

  • Introduction, what is the rationale of the study? The readers know there is a lack of studies conducted between swimmers and kayakers, but why need to conduct this study? Please include in the introduction. The conceptual framework needs to specify more clearly. Unfortunately, the current framework of this study is unclear.

The following periods have been added to the introduction: “Both kayakers and swimmers indeed, require high level of power and use similar muscle groups for propulsion in water. However, the difference in drag between the boat-paddle system and the swimmers body requires different power expressions. Elite athletes are very specialized, the inclusion of PR and PU exercises in specific resistance training programs however, is very common among both kayakers and swimmers.”

  • Methods – please include sample size calculation.

The following period has been included: “The estimated sample size was 24 to detect a between-group difference of 150w in both PR and PU.”

  • What events are the elite swimmers competing? Same for kayakers?

The following information have been included in the Participants section: “competing in front-crawl short- and middle- distance events (100-400 m) ……. competing in K1, K2 and K4 short- and middle- distance events (500-1000 m).”

  • The area under the curve (AUC) in the abstract, but in Methods author used “area under the force-power curve (AUC)”

Amended

  • In method: tested 2 sessions separated by 48 hours. Please support with reference, is it sufficient to fully rest? How was the randomization between PR and PU occur?

The following information has been included in the section: “…was randomized for each participant by flipping a coin.” The authors believe that 48 hours between maximal strength assessments may be enough since the participants were accustumed to this kind of evaluation. In addition, several studies used 48 h between maximal strength and power assessments (Wang et al. ,2017. Evaluating upper-body strength and power from a single test: The ballistic push-up. JSCR 31(5), 1338-1345.).

  • The study involves human participants, please attached the Human ethics certificate and code.

Please finda ttached the human ethics certificate.

  • Why did researchers not use better measurement like the isokinetic machine to measure power? Please cite the procedure.

In the present study bench press and the bench pull test were performed because the authors believe that reproducing the usual assessments performed by both swimmers and kayakers may improve the relevance of the study and the potential interest of strength and conditioning experts. In addition, familiarization protocols are not always suitable for elite athletes.

  • Section “2.4 statistics” is confusing. “if the assumption of sphericity was violated…..”. This sentence is missing the main statistical analysis. Did researchers use mixed factorial ANOVA or repeated measured ANOVA? Assumption of sphericity normally checked under this Analysis (Repeated measured). The statistical software used in this analysis is not mentioned in the text.

In the first version of the manuscript, one factorial ANOVA was performed. However, as suggested by the reviewers, independent sample t tests were performed. The periods:  “If the assumption of sphericity was violated, a Greenhouse-Geisser correction was applied The partial eta squared statistic was reported as the effect size (ES), and according to Stevens (20), 0.01, 0.06, and 0.14 represents small, medium, and large effect sizes, respectively” have been removed. P values have been changed in the manuscript. The following reference has been removed : “20. Stevens, JP. Applied Multivariate Statistics for the Social Sciences (Chapter 4). New York, NY: Routledge, 2012. pp. 169.”

  • Is there any interaction effect and time effect (40, 60, 80)? This was not assessed in the study.

Since t tests have been run, no interactions between the different loads in PR and PU have been checked. However, power produced at the different loads in both PU (PU40, PU60 and PU80) and PR (PR40, PR60, PR80) were compared between the two groups.

  • What is the demographic information of the participants? Please include in a Table and analyze the differences, rather than just descriptive in the Participants section. Besides, suggest adding BMI and years of experiences competing at the international level. Is it junior international or senior? Much unknown information.

BMI has been calculated and compared between the two groups. In addition, body mass and height have been recalculated in Kayak group. Independent t test did not show any significant difference between the groups for athropometric parameters. The following period has been included in the manuscript: “No significant group differences (p>0.05) were detected for any anthropometric parameter.”

  • Similar to Table 1, suggest to analyze the difference and show the effect size and p-values between swimmers and kayakers.

p values have been included in the Table.

  • Sampling method and study design were not mentioned in the manuscript.

The following period has been included in the section: “Participants of both groups were members of the Italian Senior National Team and selected for the training camps of their federation”

Reference: Only one 2020, and two 2018, others are fairly old references. Besides, please include doi.

DOI have been included in the references.

  • The authors' contribution was also imbalance, seemed like S.B did the entire experiment and writing.

The contribution of GG and MC was very important since they deeply review the manuscript. In addition MC analyzed the data.

  • I think this manuscript is not suitable for IJERPH, suggest submitting it under “Sports” MDPI journal.

Unfortunately, the manuscript is not ready for publication in this form.  

Thank You

Reviewer 2 Report

In the manuscript entitled “A comparison between elite swimmers and kayakers on upper body push and pull strength and power performance”, the authors compared outcome variables from bench press and bench pull exercises between the two groups. The sample quality is impressive, and the results are quite interesting. This study would give a good insight into science and practice of aquatic sports. That being said, there are two major concerns that the reviewer would like the authors to consider.

The first concern is the variable “AUC (the area under force-power curve”). The reviewer does not understand this quantity. Firstly, the power plot as a function of force would not form a single line, but it would provide rather a circle-like plot (since there should be several power values plotted on a single force value) – and therefore it is not possible to calculate the area “under the curve”. Secondly, even if that were possible, the area would be a quantity representing the power integrated by force, with the unit of W. N (Watt Newton). It is hard to understand what this variable means and what it tells readers. Furthermore, the authors put the unit for this variable as (a.u.). As far as the reviewer knows, this unit stands for “astronomical unit (1 a.u. = 149 597 870 700 m). Surely this was not what the authors calculated. Please include more information on what it is and why it is necessary, or simply delete this variable from the manuscript. The main outcomes are 1RM ratio and POW ratio, and the reviewer thinks AUC is not necessarily a result that should be presented here.

The second concern is the statistics. The authors describe that “If the assumption of sphericity was violated, a Greenhouse-Geisser correction was applied”. This does not sound right to the reviewer. Sphericity (with Greenhouse-Geisser correction) is the assumption one should check in a repeated-measures design, and not in independent ANOVAs. In an independent design such as the current study, homogeneity of variances should be checked. Please check the statistical procedure once again and redo the analysis if any mistakes are to be found. Relating to the statistics, why did the authors run one-way ANOVA and not an independent t-test? ANOVA would perfectly make sense if the authors applied a two-way mixed ANOVA (group as an independent factor and % 1RM as a repeated factor). However, it was not the case, as the authors described the process as a one-factor analysis of variance and no %1RM effect or interaction were reported in the result. Utilising a one-way ANOVA for two groups is not a wrong process, but the reviewer is curious why the authors chose to do so.

Specific comments (There are no line numbers presented in the manuscript, so only the page number is described when necessary)

Page2. Last three sentences in the introduction. Please change “upper body strength” to more specific words. Also, in the last sentence, change “may be found in the two groups of athletes” to “exist between the two groups of athletes”.

The reviewer suggests changing the abbreviation for the power to a more standard one, such as P (and peak power for Ppeak)

In the discussion (page 6), the authors introduce the study by McKean et al. where 1RM ratio of 0.76 (elite kayakers) was reported. On the other hand, in the introduction, the authors refer to another study that reported the quantity of 0.96 for elite kayakers. Given the large variability between the studies (including the current study in which the outcome was 0.82), some consideration and discussion on this inconsistency would be necessary.

Again, in page 6, the authors discuss differences between swimmers, kayakers, and athletes in other sports. The discussion is based on the absolute value, but the reviewer thinks readers would be interested in relative terms as well (such as % body mass). For example, given that rugby players are likely much heavier, perhaps swimmers and kayakers are as strong as them when compared using relative 1 RM?

Page 6, second last paragraph. “propelling efficiency and water perception seem to be key factors for swimming performance.” Please include references that support this statement.

Page 6, second last paragraph. The authors discuss the difference between swimming and kayaking using the stroking time (about 0.47 s) and stress the potential difference in speed and its effect on the importance of the power requirement. However, the reviewer disagrees with this. In maximum effort swimming, the cycle time is about 1 s, and swimmers perform two arm strokes in that short period of time, which gives the one-arm underwater time of approximately 0.5 s, which is almost the same as the potential stroking time in kayaking. Of course, the kayak boat propels much faster than swimming human body, so the speed of the oar relative to the boat would be faster than the relative hand speed in swimming. However, the authors should not ignore the fact that the end-effector speed is largely affected by the length of the arm. The distance between the kayak paddle to the shoulder and the hand to the shoulder is obviously very different – and therefore the shoulder joint angular velocity might not be that different between swimming and kayaking. Rather, the reviewer thinks the critical factor that affects the power is the force rather than joint angular velocity, as the end-effector size (puddles and hands) are very different between the conditions. Perhaps studies by Tsunokawa et al. (2017; 2019) who investigated the effect of hand paddles on the fluid forces during swimming might be useful for the discussion.

Author Response

The authors would like to thank the reviewers for their time and for their suggestions in the attempt to improve the overall quality of the manuscript. In the manuscript, changes have been written in red while removed periods have been highlighted in yellow.

Rew2

In the manuscript entitled “A comparison between elite swimmers and kayakers on upper body push and pull strength and power performance”, the authors compared outcome variables from bench press and bench pull exercises between the two groups. The sample quality is impressive, and the results are quite interesting. This study would give a good insight into science and practice of aquatic sports. That being said, there are two major concerns that the reviewer would like the authors to consider.

The first concern is the variable “AUC (the area under force-power curve”). The reviewer does not understand this quantity. Firstly, the power plot as a function of force would not form a single line, but it would provide rather a circle-like plot (since there should be several power values plotted on a single force value) – and therefore it is not possible to calculate the area “under the curve”. Secondly, even if that were possible, the area would be a quantity representing the power integrated by force, with the unit of W. N (Watt Newton). It is hard to understand what this variable means and what it tells readers. Furthermore, the authors put the unit for this variable as (a.u.). As far as the reviewer knows, this unit stands for “astronomical unit (1 a.u. = 149 597 870 700 m). Surely this was not what the authors calculated. Please include more information on what it is and why it is necessary, or simply delete this variable from the manuscript. The main outcomes are 1RM ratio and POW ratio, and the reviewer thinks AUC is not necessarily a result that should be presented here.

The authors understand that AUC may be quite difficult to figure out. These parameter however has been extensively used in researches about resistance training (Bartolomei et al. (2018). Comparison between bench press throw and ballistic push-up tests to assess upper-body power in trained individuals. The Journal of Strength & Conditioning Research32(6), 1503-1510; Bartolomei et al. (2017). The influence of isometric preload on power expressed during bench press in strength-trained men. European journal of sport science17(2), 195-199.). The authors believe that AUC may be important to give a parameter of power produced by athlete at the different loads. AUC indeed, put together the power produced by each athlete at the different loads. The following reference has been included in the manuscript: “20. Bartolomei, S.; Fukuda, D.H.; Hoffman, J.R.; Stout, J.R.; Merni, F. 2017. The influence of isometric preload on power expressed during bench press in strength-trained men. European journal of sport science, 2017, 17(2), 195-199”. The unit of measure is expressed in a.u. (arbitrary units). However, in Figure 1 and Figure 2 a.u. has been replaced with Power (w). That was a mistake. In addition, power expressed at the different loads in both PR and PU have been compared between the two groups and presented in the “Results” section.

The second concern is the statistics. The authors describe that “If the assumption of sphericity was violated, a Greenhouse-Geisser correction was applied”. This does not sound right to the reviewer. Sphericity (with Greenhouse-Geisser correction) is the assumption one should check in a repeated-measures design, and not in independent ANOVAs. In an independent design such as the current study, homogeneity of variances should be checked. Please check the statistical procedure once again and redo the analysis if any mistakes are to be found. Relating to the statistics, why did the authors run one-way ANOVA and not an independent t-test? ANOVA would perfectly make sense if the authors applied a two-way mixed ANOVA (group as an independent factor and % 1RM as a repeated factor). However, it was not the case, as the authors described the process as a one-factor analysis of variance and no %1RM effect or interaction were reported in the result. Utilising a one-way ANOVA for two groups is not a wrong process, but the reviewer is curious why the authors chose to do so.

The authors agree with the reviewer. Since a one-way anova was run, there’s no need for greenhouse -Geisser correction. ANOVA however, has been replaced with indepedent sample t tests. Thus, p values were checked and corrected in the manuscript. p values were also included in Table 1.

Specific comments (There are no line numbers presented in the manuscript, so only the page number is described when necessary)

Page2. Last three sentences in the introduction. Please change “upper body strength” to more specific words. Also, in the last sentence, change “may be found in the two groups of athletes” to “exist between the two groups of athletes”.

Amended. “Upper body” has been removed from the last three periods.

The reviewer suggests changing the abbreviation for the power to a more standard one, such as P (and peak power for Ppeak)

Amended. POW Peak has been replaced with PPeak and POW ratio has been replaced with P ratio.

In the discussion (page 6), the authors introduce the study by McKean et al. where 1RM ratio of 0.76 (elite kayakers) was reported. On the other hand, in the introduction, the authors refer to another study that reported the quantity of 0.96 for elite kayakers. Given the large variability between the studies (including the current study in which the outcome was 0.82), some consideration and discussion on this inconsistency would be necessary.

The following period has been included in the section: “Curiously, a push/pull ratio of 0.98 has been found in elite Turkish kayakers. Turkish kayakers however, may not represent the world elite. On the contrary, in the study of McKean (7) and in the present investigation, participants were world top level athletes.”

Again, in page 6, the authors discuss differences between swimmers, kayakers, and athletes in other sports. The discussion is based on the absolute value, but the reviewer thinks readers would be interested in relative terms as well (such as % body mass). For example, given that rugby players are likely much heavier, perhaps swimmers and kayakers are as strong as them when compared using relative 1 RM?

The following periods have been included to the section: “In addition, Baker and Newton (4) reported an average body mass of 94.4 kg in professional Rugby Players while the average body mass of both KA and SW participating in the present study was about 83 kg. KA and SW athletes indeed, may be stronger than other athletic populations in pull strength relative to body mass.

Page 6, second last paragraph. “propelling efficiency and water perception seem to be key factors for swimming performance.” Please include references that support this statement.

 Amended. The following reference has been included in the manuscript: “ Gatta, G., Cortesi, M., Swaine, I., & Zamparo, P. (2018). Mechanical power, thrust power and propelling efficiency: relationships with elite sprint swimming performance. Journal of Sports Sciences, 36(5), 506-512.

Page 6, second last paragraph. The authors discuss the difference between swimming and kayaking using the stroking time (about 0.47 s) and stress the potential difference in speed and its effect on the importance of the power requirement. However, the reviewer disagrees with this. In maximum effort swimming, the cycle time is about 1 s, and swimmers perform two arm strokes in that short period of time, which gives the one-arm underwater time of approximately 0.5 s, which is almost the same as the potential stroking time in kayaking. Of course, the kayak boat propels much faster than swimming human body, so the speed of the oar relative to the boat would be faster than the relative hand speed in swimming. However, the authors should not ignore the fact that the end-effector speed is largely affected by the length of the arm. The distance between the kayak paddle to the shoulder and the hand to the shoulder is obviously very different – and therefore the shoulder joint angular velocity might not be that different between swimming and kayaking. Rather, the reviewer thinks the critical factor that affects the power is the force rather than joint angular velocity, as the end-effector size (puddles and hands) are very different between the conditions. Perhaps studies by Tsunokawa et al. (2017; 2019) who investigated the effect of hand paddles on the fluid forces during swimming might be useful for the discussion.

The authors understand the reviewer’s concern. The section has been changed as follows: “The lower velocity of the swimmer is partially compensated by the absence of paddles, thus the shoulder joint angular velocity may not be different between sprint swimming and kayaking. The absence of paddles however, may reduce the need for maximal pull strength in swimming.”

Thank you.

Reviewer 3 Report

General Comments

This study compares the strength and power relationship in pull and push actions between swimmers and Kayakers. It concludes that both groups present higher strength in the pull and that the kayakers are generally more powerful and have a higher 1RM. The research is novel and interesting in an applied perspective.

However, there are some limitations in the introduction and study design that precludes the manuscript for publication in this journal. My main concerns are that the introduction does not clearly establish the need for the study and the hypotheses is not sufficiently supported by the previous parts of the in some parts of the text. Most importantly, authors use “force”, strength” and power indiscriminately. For example the title of the paper mentions strength, but the abstract mentions that goal of the study is to compare the force power curve. This should be corrected. The authors testes the load power curve and not the force, since they did not measure torque or force. This is an important question that the authors should address and change the manuscript accordingly. Furthermore, the study design should provide more detail and the statistics analysis and results should be reformulated.

Conversely, the discussion is well designed and supported by the literature, establishing reasonable training inferences from the data.

It is also my opinion that the writing could be improved since there are several grammatical errors throughout the text. Please revise the English and preferentially seek an English native to correct the grammatical errors.

Specific comments

Lines 34 and 35, please remove indeed.

Line 45 - please explain... not clear if the authors mean the force/velocity curve or load power curve... the terms force/strenght and power seem to be used in an indiscriminated manner...

Line 53 – “Despite several studies have” . insert that before have.

Line 54 – “no studies analyzed the power ratios in elite athletes”- The physiological meaning of the power ratios is not provided before.

Line 58/59 – The hyphotheses is not supported by the previous parts of the introduction.

Line 64 - please provide more data regarding the athletes, namely the average training volume, FINA points in the case of swimmers; Internacional ranking in the case of kayakers if possible.

Line 70 – speeling error in previuos

Line 71/72 – did the athletes trained their specific modality before the testing sessions? How was fatigue levels controlled?

Line 122/123 - what performance variables, please specify. I do not understand some of the statistical procedures used. What is the rationale for the Anova if you only have 2 groups?

Is the Effect size reported? And discussed? If not, it should be removed from the methods.

Table 1

please state the meaning of *.

In the Figures mark the significant differences.

Line 175/176   -  please detail

Line 221 – spelling error in Contrariwise

Line 224/225 – “To the best of our knowledge, this is the first study to investigate the 224 ratio between push and pull power in elite athletes (POW Ratio). KA athletes showed a push-pull 225 POW ratio of 0.50 while SW of 0.89.” these results are not presented in the results section.

Author Response

The authors would like to thank the reviewers for their time and for their suggestions in the attempt to improve the overall quality of the manuscript. In the manuscript, changes have been written in red while removed periods have been highlighted in yellow.

Rev3

This study compares the strength and power relationship in pull and push actions between swimmers and Kayakers. It concludes that both groups present higher strength in the pull and that the kayakers are generally more powerful and have a higher 1RM. The research is novel and interesting in an applied perspective.

However, there are some limitations in the introduction and study design that precludes the manuscript for publication in this journal. My main concerns are that the introduction does not clearly establish the need for the study and the hypotheses is not sufficiently supported by the previous parts of the in some parts of the text. Most importantly, authors use “force”, strength” and power indiscriminately. For example the title of the paper mentions strength, but the abstract mentions that goal of the study is to compare the force power curve. This should be corrected. The authors testes the load power curve and not the force, since they did not measure torque or force. This is an important question that the authors should address and change the manuscript accordingly. Furthermore, the study design should provide more detail and the statistics analysis and results should be reformulated.

Force-power has been replaced with load-power in the manuscript.  Study design has been improved and further informations have been included. Statistical analysis has been reformulated. The following paragraph has been included in the introduction: “Both kayakers and swimmers indeed, require high level of power and use similar muscle groups for propulsion in water. However, the difference in drag between the boat-paddle system and the swimmers body requires different power expressions. Elite athletes are very specialized, the inclusion of PR and PU exercises in specific resistance training programs however, is very common among both kayakers and swimmers.

Conversely, the discussion is well designed and supported by the literature, establishing reasonable training inferences from the data.

It is also my opinion that the writing could be improved since there are several grammatical errors throughout the text. Please revise the English and preferentially seek an English native to correct the grammatical errors.

Specific comments

Lines 34 and 35, please remove indeed.

Amended

Line 45 - please explain... not clear if the authors mean the force/velocity curve or load power curve... the terms force/strenght and power seem to be used in an indiscriminated manner...

The period has been changed as follows: “Different typical strength and power expressions”

Line 53 – “Despite several studies have” . insert that before have.

Amended

Line 54 – “no studies analyzed the power ratios in elite athletes”- The physiological meaning of the power ratios is not provided before.

The following period has been included in the manuscript: “This parameter may give additional information about the relationships between anterior and posterior shoulder muscle groups from a more sport-specific perspective.”

Line 58/59 – The hyphotheses is not supported by the previous parts of the introduction.

The following periods have been included in the section: “Both kayakers and swimmers indeed, require high level of power and use similar muscle groups for propulsion in water. However, the difference in drag between the boat-paddle system and the swimmers body requires different power expressions. Elite athletes are very specialized, the inclusion of PR and PU exercises in specific resistance training programs however, is very common among both kayakers and swimmers.”

Line 64 - please provide more data regarding the athletes, namely the average training volume, FINA points in the case of swimmers; Internacional ranking in the case of kayakers if possible.

As suggested by the reviewer, FINA points and Kayak average world ranking have been included.

Line 70 – speeling error in previuos

Amended

Line 71/72 – did the athletes trained their specific modality before the testing sessions? How was fatigue levels controlled?

Athletes participating in the present study were not allowed to train in the 48 h before each assessment. Testing sessions were performed during the off season, the only moment of the year in which they could stay 2 days without sport specific training.

Line 122/123 - what performance variables, please specify. I do not understand some of the statistical procedures used. What is the rationale for the Anova if you only have 2 groups?

ANOVA has been replaced with independent sample t test.

Is the Effect size reported? And discussed? If not, it should be removed from the methods.

Effect size has been removed by methods.

Table 1

please state the meaning of *.

P values have been reported in Table 1.

In the Figures mark the significant differences.

Amended

Line 175/176   -  please detail

The following period has been included in the discussion: “The need for high levels of maximal strength and power in KA, mainly involves shoulder back muscles.”

Line 221 – spelling error in Contrariwise

Amended

Line 224/225 – “To the best of our knowledge, this is the first study to investigate the 224 ratio between push and pull power in elite athletes (POW Ratio). KA athletes showed a push-pull 225 POW ratio of 0.50 while SW of 0.89.” these results are not presented in the results section.

The authors understand the reviewer’s query. These results however, are reported in Table 1.

Thank you.

Round 2

Reviewer 1 Report

Please attach the actual sample size calculation and Human ethics research approval. 

Thanks.

Author Response

Please attach the actual sample size calculation and Human ethics research approval. 

Please find attached the sample size calculation and the human ethics approval.

Sample size calculation.

Assuming a pooled standard deviation of 90 w in power assessments [as previously found by Bartolomei et al. Effect of lower body resistance training on upper body strangth adaptation in resistance trained men. JSCR 32(1), 13-18,2016] the study would require a sample size of 12 for each group to achieve a power of 80% and a level of significance of 5% for detecting a true difference in means between the groups of 100 w.

Thank you.

Reviewer 3 Report

The revised version of the paper adequately addresses my comments. 

I commend the authors for studying elite athletes. 

I only want to point that although the authors mention that the effect size was removed from the methodology since it was not in the results or discussion, it is still in lines 137/138. 

Author Response

The revised version of the paper adequately addresses my comments. 

I commend the authors for studying elite athletes. 

Thank you

I only want to point that although the authors mention that the effect size was removed from the methodology since it was not in the results or discussion, it is still in lines 137/138. 

Effect size has been removed in line 137-138

Thank you